# Ground deformation reveals the scale-invariant conduit dynamics driving explosive basaltic eruptions

M. Ripepe [1 ✉], G. Lacanna[1], M. Pistolesi [2], M. C. Silengo[1], A. Aiuppa [3], M. Laiolo [1,4], F. Massimetti[1], L. Innocenti[1], M. Della Schiava[1], M. Bitetto [3], F. P. La Monica [1], T. Nishimura[5], M. Rosi[2], D. Mangione [6], A. Ricciardi[6], R. Genco[1], D. Coppola [4], E. Marchetti[1] & D. Delle Donne[7]

The mild activity of basaltic volcanoes is punctuated by violent explosive eruptions that occur without obvious precursors. Modelling the source processes of these sudden blasts is challenging. Here, we use two decades of ground deformation (tilt) records from Stromboli volcano to shed light, with unprecedented detail, on the short-term (minute-scale) conduit processes that drive such violent volcanic eruptions. We find that explosive eruptions, with source parameters spanning seven orders of magnitude, all share a common pre-blast ground inflation trend. We explain this exponential inflation using a model in which pressure build-up is caused by the rapid expansion of volatile-rich magma rising from depth into a shallow (<400 m) resident magma conduit. We show that the duration and amplitude of this inflation trend scales with the eruption magnitude, indicating that the explosive dynamics obey the same (scale-invariant) conduit process. This scale-invariance of pre-explosion ground deformation may usher in a new era of short-term eruption forecasting.

[1] Dipartimento di Scienze della Terra, Università di Firenze, Firenze, Italy. [2] Dipartimento di Scienze della Terra, Università di Pisa, Pisa, Italy. [3] Dipartimento di Scienze della Terra e del Mare, Università di Palermo, Palermo, Italy. [4] Dipartimento di Scienze della Terra, Università di Torino, Torino, Italy. [5] Department of Geophysics, Graduate School of Science, Tohoku University, Sendai, Japan. [6] Dipartimento di Protezione Civile, Roma, Italy. [7] Istituto Nazionale di Geofisica e Vulcanologia, Osservatorio Vesuviano, Napoli, Italy. ✉email: maurizio.ripepe@unifi.it

The high death toll (>1100) claimed by explosive volcanic eruptions in the last decade[1–3] dramatically highlights that, in spite of enormous progresses in volcano monitoring[4], explosive volcanic blasts can still occur suddenly and without any recognized precursors. Unsuccessful forecasts are a consequence of our incomplete understanding of the magmatic processes driving these explosions, which, unlike dike intrusions[5–7] or magma ascending from deep reservoirs[8,9], act over timescales of only seconds to minutes[10–16].

Such sudden explosive eruptions are known to be associated with inflation/deflation cycles[13–15] caused by recharge/discharge of the feeding magma column. However, in spite of the recently much improved accuracy ($10^{-9}$ rad) and temporal resolution (1 s) of tilt measurements, the intrinsic challenges in capturing and identifying the subtle ground deformation associated with explosive eruptions[10–16] limit the use of these signals to trigger warnings in real-time alert systems[17,18].

Violent (volcanic explosivity index, VEI ≤ 3) and unexpected explosive eruptions also occasionally occur at open-conduit volcanoes, whose "regular" activity primarily takes the form of passive degassing and far milder, low-intensity (i.e. Strombolian) explosive activity[19,20]. Such "paroxysmal" explosions represent a real threat for local inhabitants, scientists and visitors due to the relatively large dispersal of ballistics and pyroclastic flows[21]; this activity further illustrates the challenges in managing volcanic risk[22] in densely inhabited regions that are also prone to volcano tourism[23].

Some of the most spectacular and violent "basaltic" explosions occur at Stromboli, a volcano in Southern Italy globally renowned for its persistent Strombolian activity[24,25]. Stromboli's regular, mild explosive activity is occasionally interrupted or accompanied by lava effusions[26] inside the gravitationally unstable Sciara del Fuoco scar and, even more critically, by violent explosive eruptions referred to as "major explosions" and "paroxysms"[24,25,27]. In contrast to regular explosions, characterized by eruptive masses of ~$10^3$ kg ejected to heights of ~150 m at a rate of $10^2$ kg/s[28], major explosions and paroxysms typically erupt $10^5$–$10^8$ kg of tephra at rates reaching up to $10^7$ kg/s (refs. [29,30]) and feed 1500–5000-m-high convective columns. These types of eruptions are also associated with distinct magma properties. Whereas regular Strombolian explosions are fed by volatile-poor (<1 wt% $H_2O$), high-density (2700 kg/m³), high-viscosity ($1$–$4 \times 10^4$ Pa s), crystal-rich magma stored in shallow (≤3 km) conduits[31,32], paroxysmal and major explosions are unique in that they erupt gas-rich (3 wt% $H_2O$, 2 wt% $CO_2$; refs. [33,34]), low density (2500 kg/m³), low viscosity (<100 Pa s), and crystal-poor magma that rapidly ascends from a deep-seated (~7–10 km depth) reservoir[31,34].

The most recent paroxysmal explosions at Stromboli occurred on 3 July and 28 August 2019 (Supplementary Note 1). Both events resulted in several kilometre-high convective columns, produced showers of ash, lapilli, and bombs, and generated tsunamigenic pyroclastic flows along the Sciara del Fuoco[35]. The 3-July event tragically caused the loss of one life; however, if the explosion had occurred only a few hours later, hundreds of tourists visiting the summit would have been severely affected with even greater tragedy.

Here, we analyze the ground tilt measured during Stromboli's explosive eruptions in 2019 to identify a systematic pre-blast inflation trend, which we interpret as the response of the volcanic edifice to the pressure growth induced by the rapid gas expansion inside the conduit. We show that this exponential inflation pattern is common to a wide range of explosive events spanning almost five orders of magnitudes in intensity and associated volumes. This scale-invariant ground deformation opens new perspectives to explain explosive dynamics, and paves the way to improved volcano monitoring, with obvious benefits for hazard assessment and mitigation.

## Results

**Ground deformation analysis and modelling.** We analyse the ground tilt records associated with the 2019 and earlier (5 April 2003 and 15 March 2007) paroxysmal eruptions at Stromboli (Fig. 1). The 3 July and 28 August 2019 violent explosions both generated considerable ground deformation of 14 and 9 µrad, respectively, recorded by tiltmeters as a gradual inflation of the volcano edifice starting almost 10 min before the explosive onset (Figs. 1 and 2; Supplementary Video 1). In addition to the use of tiltmeters, we also derive ground deformation measurements from seismic broadband stations[12,13] within a permanent network (Methods and Supplementary Fig. 1). We find that the deformation amplitude rapidly decays with distance from the craters (Fig. 1), suggesting a shallow source origin. We locate the source of deformation by assuming a cylindrical open conduit[36] of radius $a$, where the tilt at the ground surface, $t$, is:

$$\tau = G \frac{a^2}{\mu} \Delta P \tag{1}$$

defined by the Green's function $G$ and rigidity $\mu$ of the country rock (see Methods). The inflation source is constrained (Fig. 1) to the upper portion of the magmatic conduit(s) at a depth much shallower than the estimated depth of deformation of the 15 March

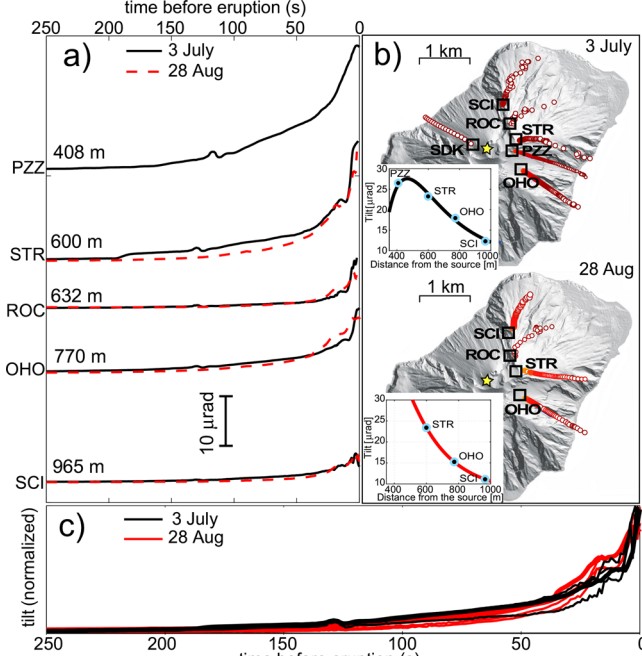

**Fig. 1 Ground tilt data for the 3 July and 28 August 2019 paroxysms. a** Radial tilts recorded by four broadband seismic stations (PZZ, STR, ROC, and SCI) and the borehole tiltmeter (OHO) of the permanent network (Supplementary Fig. 1) 250 s before the onsets of the 3 July 2019 (black line) and 28 August 2019 (red dashed) paroxysms within a distance of 408–965 m from the vent. **b** Tilt trajectories associated with the waveforms shown in **a** showing an almost radial pattern pointing to the crater area for both paroxysms. The best-fit solutions (open-conduit deformation model[36]) between the theoretical tilt pattern and observations are shown in the insets and are consistent with volume changes of $4.9 \times 10^4$ m³ and $3.2 \times 10^4$ m³ in the upper 380 and 240 m a.s.l. below the vent for the 3 July 2019 and 28 August 2019 paroxysms, respectively. **c** Normalized radial tilt waveforms corresponding to those shown in **a** associated with the two paroxysms illustrating how the amplitude ratio between stations remained the same during the 250 s before the onset of the paroxysm, suggesting a stationary shallow source. Due to calibration issues at seismic station SDK, the absolute tilt could not be derived; hence, its data were not used in the modelling.

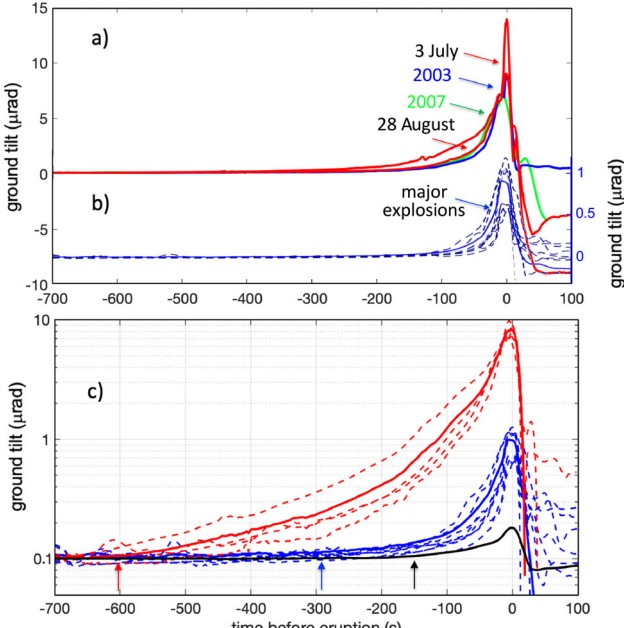

**Fig. 2 Comparing tilt for regular, major, and paroxysms eruptions.**
**a** Ground deformation associated with the paroxysms of July and August 2019 (red lines), April 2003 (blue line) and March 2007 (green line) in comparison with **b** the ground tilt recorded during major explosions (dashed blue lines). Note that the ground tilt for the paroxysms and major eruptions have different scales (one order of magnitude) but similar deformation rate. **c** The different scale of the deformation rate is better evidenced on a semilog scale where the stacking deformation among the four (dashed red lines) paroxysms (bold red line) is represented together 530 with the stacked deformations of major (bold blue line) and regular (black line) explosions[13]. The amplitude of the tilt increases by almost one order of magnitude from regular (~0.1 μrad) to major explosions (~1 μrad) and again to paroxysms (~10 μrad). Note how the exponential rate becomes progressively more evident when the explosive activity becomes more intense and how ground inflation becomes visible at ~150 s (black arrow) before a regular explosion, ~300 s (blue arrow) before a major explosion and more than 600 s (red arrow) before a paroxysm.

2007 paroxysm[37] but well in agreement with the source location determined by very-long-period seismicity[38] and with that of the ground displacement measured by 1 Hz GPS stations during the 5 April 2003 paroxysm[39]. The best solution for the 3 July 2019 paroxysm (best fit: 0.98) is compatible with the inflation of the uppermost 380 m of the conduit (from 750 to 370 m above sea level, a.s.l.) attributed to a volumetric expansion of $4.9 \times 10^4 \, m^3$, whereas the 28 August 2019 eruption (best fit: 0.99) implies the deformation of a smaller conduit portion (240 m; from 720 to 480 m a.s.l.) and a volumetric expansion of $3.2 \times 10^4 \, m^3$. The constant ratio between the ground tilt measurements at different stations (Fig. 1) implies that the source of inflation is stable at the same position and does not move over time. For a shallow magma conduit with radius $a = 95$ m (ref. [26]) and a rigidity $\mu$ of $1.3 \times 10^9$ Pa s (ref. [13]), the ground tilts for the 3 July and 28 August 2019 paroxysms correspond to conduit overpressures of 5.9 and 6.1 MPa, respectively. Explosions occurred at the end of the inflation phase and were followed by deflation of the same amplitude but lasting only 50–70 s.

**Ground inflation and explosive dynamics**. This inflation-deflation pattern is not unique to paroxysms but rather is a general characteristic feature of the explosive process at Stromboli (Fig. 2). Regular Strombolian activity manifests in a rhythmic sequence of recharge/discharge cycles of ~0.1 μrad amplitude each[13], in which explosions are preceded by an ~150 s long gradual inflation of the ground, followed by a rapid deflation phase lasting ~20 s (Fig. 2b) as the conduit contracts in response to the ejection of gas and magma from the vent[13]. Similar inflation/deflation cycles are associated with larger-scale explosive events (i.e. major explosions and paroxysms), albeit with far larger amplitudes. Over the last 17 years (2003–2020), 39 major explosions and four paroxysms were recorded interspersed within the prevailing regular activity. All of these events were associated with ground inflation preceding the explosive onset by several minutes (>300 s; Fig. 2). When the stacked ground deformation recorded during regular Strombolian activity is compared with the stacked deformation recorded during the 39 major explosions and during the four paroxysms, the inflation amplitude and duration both scale with the magnitude of the explosion (Fig. 2b): the larger the explosion is, the earlier the onset and the larger the amplitude of the signal. At a distance of 800 m from the craters, paroxysms generate substantial ground inflation of ~10 μrad starting more than 600 s before the explosive onset (Fig. 2c), whereas major explosions give rise to long ground inflation (~300 s) of ~0.8 μrad (averaged over the 39 major explosions), one order of magnitude smaller than that generated by paroxysms and one order of magnitude larger than that of regular explosions (~0.1 μrad; Fig. 2b). In addition, the amplitude of ground tilt scales remarkably with the tephra volume erupted during different explosive activities, as obtained from deposit studies (Fig. 3), suggesting a strong relationship between ground deformation and explosive dynamics.

## Discussion

Relative to those of regular Strombolian explosive activity, the distinct magnitude/intensity and magma chemistry of Stromboli's major and paroxysmal eruptions have been taken as evidence of diverse conduit processes and/or different dynamics between deep and shallow magmatic systems[27,31,34]. However, once the ground deformation is normalized, regular, major and paroxysmal explosive eruptions follow identical inflation trends (Fig. 4a). The rate at which the ground inflates obeys the same exponential trend, indicating that regular activity, major explosions, and paroxysms, despite the activation of different magma reservoirs, share scale invariant, shallow conduit dynamics that evolve similarly in time but with different durations and thus different depths of the source nucleation process. We conclude that, while the amplitude of the pre-explosion deformation scales with the eruption magnitude, the temporal deformation pattern is independent of the depth at which the process started and of the intensity of the ensuing explosive event, indicating a repetitive shallow conduit process.

We characterize the pre-paroxysm, in-conduit process (assuming its pre-onset ground inflation phase is produced by a pressure increase) as vesiculated magma rising from depth[40,41] expanding in the shallow conduit. Our model is derived from independent petrological evidence that attributes paroxysms to the rapid injection of volatile-rich deep magma into higher-viscosity, crystal-rich and degassed conduit magma[31–34]. However, the differences in the gas and crystal contents (and, hence, in the viscosity) between the two magma types are not explicitly taken into account in our model. We nevertheless argue that the low viscosity of mafic, gas-rich magma rising from depth favours its rapid upward migration and interaction with denser, degassed magmas residing in the shallow conduit, which can act as a viscous plug promoting resistance to conduit flow[42,43].

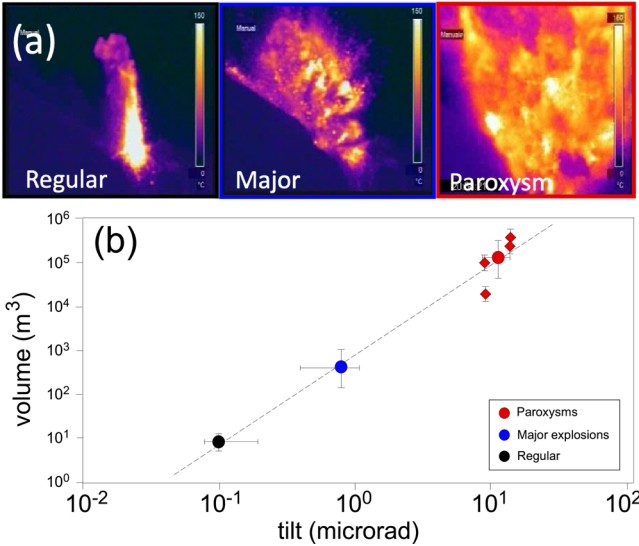

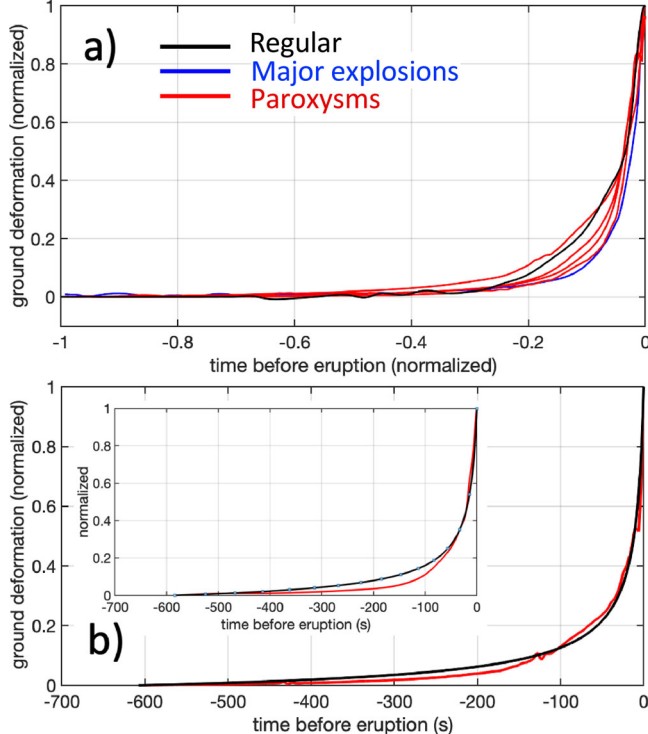

**Fig. 3 Tephra volumes and ground tilt. a** Thermal images taken by the same camera located ~450 m from the craters showing the different scale of intensity of the explosive eruptions at Stromboli based on the temperature, column height and product dispersal. The images show examples of regular Strombolian activity (left frame), the major explosion that occurred on 08 November 2009 (central frame), and the 15 March 2007 paroxysm (right frame). **b** Logarithmic plot of the volume of erupted product and ground tilt for regular Strombolian activity, major explosions and paroxysms. Tilt error bars for Strombolian activity are based on >10,000 observed events, and those for major explosions are estimated based on the 39 major eruptions that occurred from 2005 to 2019. The average volume for paroxysms (circle) is based on measurements (diamonds) from the 2003 and 2007 events (data are from refs. [20,25]), whereas the volumes of the 2019 paroxysms are estimated on the single exponential decay law from the loading per unit area of fallout deposits versus isomass area plots[54].

**Fig. 4 A Scale-invariant of ground deformation pattern. a** Normalized ground tilts of the inflation events recorded during the four paroxysms (red) and the stacking of 39 major (blue) and 2000 regular (black) Strombolian explosions. Inflations (represented as a function of the normalized time before the onset of the eruption) show a very similar exponential deformation trend with a best-fit ranging between 0.89 and 0.99. These high best-fit values suggest that in spite of the different amounts of energy involved (Fig. 3), the explosive process is controlled by the same conduit dynamics. **b** The exponential trend of the inflation remarkably resembles the expansion of the gas in magma by magmastatic decompression (black line). In the case of the 3-July paroxysm (red line), the best fit (0.99) is reached using Eq. (2) for the 610 s long expansion of a vesiculated magma with a high volumetric expansion ratio of $V/Vo = 15.9$. In the inset, the expansion model calculated for the 28 August paroxysm shows a best fit of 0.98 for a volumetric expansion ratio of $V/Vo = 12$ and an expansion duration of 590 s.

In our conduit expansion model (Fig. 5), gas expands in response to magmastatic decompression, and the expansion process accelerates as the magma ascends, forcing the head of the magma column upward in the conduit. Ground inflation can thus be interpreted as resulting from the overpressure caused by magma expansion and ascension in the conduit at the base of the viscous crystal-rich layer[42,44]. Shear stress caused by upward magma migration has been found[40] to account for only ~16% of volcano deformation and is thus neglected.

If magma ascent is controlled by the expansion of a gas front in a viscous fluid[40,41], the rate at which the ground deforms in the topmost section of the conduit will increase following an exponential trend as defined by Stokes's law[40]:

$$\frac{dz}{dt} = \frac{2gR^2(t)}{9\eta}\left[\rho - \rho_g(t)\right] \quad (2)$$

where $z$ is the depth at which the gas with density $rg$ is expanding in the magma batch, $R$ is the effective gas radius, $\rho$ and $\eta$ are the bulk density and viscosity, respectively, of the surrounding magma, and $g$ is the acceleration due to gravity. Assuming ideal gas behaviour and that gas expansion starts at depth $zo$, the ideal gas law ($P_oV_o = P(t)V(t)$) can be rearranged to show that the initial gas volume ($V_o = \frac{4}{3}\pi R_o^3$) in the magma batch will grow as a function of the depth $z(t)$ upon ascending:

$$R(t) = R_o\left(\frac{Z_o}{Z(t)}\right)^{\frac{1}{3}} \quad (3)$$

The rate at which the ground deforms is thus controlled by the ascent velocity of the expanding gas-rich magma batch into the conduit (Eq. 2). Neglecting the gas density $rg$ in Eq. (2) and integrating using Eq. (3), between the initial ($zo$) and final ($zi$) positions of gas expansion (Fig. 5c), it is possible to relate the duration of inflation, $tb$, to the initial depth ($zo$) and radius ($Ro$) of the rising magma batch:

$$t_b = c\frac{9\eta}{2\rho g}\frac{z_o}{R_0^2}\left(1 - z_o^{-\frac{5}{3}}z_i^{\frac{5}{3}}\right) \quad (4)$$

where $c = 3/5$ represents the integration coefficient. Equation (4) implies that, for a given magma density and viscosity, if fragmentation occurs at the surface ($zi = 0$), the time required for the gas-rich magma to reach the surface will depend on the ratio $z_o/R_0^2$.

Using a viscosity of $\eta = 10^2$ Pa s and a density of $\rho = 2500$ kg m$^{-3}$ for crystal-poor magma[31], we calculate (Eq. 2) the gas expansion rate for all possible combinations of the ratio $z_o/R_0^2$. It is worth noting that, although the model equations (Eqs. 2 and 3) are written in terms of $R/Ro$, we prefer to express our solutions in terms of the volumetric expansion in the conduit, $V/Vo$ (the equivalent

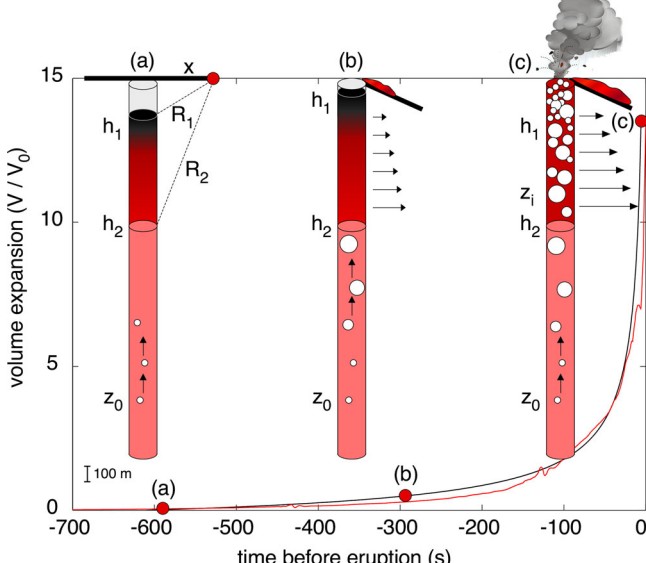

**Fig. 5 Ground deformation and conduit dynamics.** Conceptual model of the pre-eruptive process causing ground inflation lasting 610 s (red line) before the eruptive onset on 3 July 2019 and **a** the geometrical parameters used to locate the source with the open-conduit model36. In the initial phase (**a**), gas rising from depth ($z_o$) expands, forcing the magma column towards the surface (**b**) and leading to a precursory emission of lava outside the vent. Ground deformation is caused by an increase in magmastatic pressure and/or pressurization at the base ($h2$) of the viscous degassed magma at the top of the magma column ($h1$ and $h2$) in response to the exponential growth (black line) of the gas (Eq. 2). **c** When the pressure induced by the highly vesiculated magma overcomes the tensile strength of the viscous magma mush[42, 44], fragmentation occurs ($z_i$).

volumetric ratio). Assuming the expansion time $tb$ coincides with the total duration of tilt inflation (windowed from 540 to 720 s, from the duration of the pre-explosion inflation; Figs. 1 and 5), we fit the normalized inflation patterns measured by the tiltmeter with the normalized volumetric expansion ratio $V/Vo$ (considering $V/Vo = (Ro/R)^3$, Eqs. 2 and 3). The best fit (0.99) between the inflation rate recorded during the 3 July 2019 paroxysm and the model (Fig. 4b) is found for a volumetric expansion ratio $V/Vo$ of 15.8 occurring over an expansion time $tb$ of 610 s. A similar solution (expansion time $tb$ of 590 s for a volumetric expansion ratio $V/Vo$ of 12) is derived for the 28 August 2019 eruption (best fit: 0.98). Considering that expansion ends at the explosive onset, (i.e. assuming $z_i$ is the fragmentation level in the conduit), for a given $V/Vo$ ratio, the starting depth $z_o$ for the expansion of the gas-magma mixture, $z_o = (z_i + zatm)_+ + -zatm$, will depend!on $z_i$ and on the atmospheric pressure ($zatm = Patm/rg$).

Unfortunately, the fragmentation depth ($z_i$) is unconstrained for the 2019 explosions. If fragmentation was to occur at the surface ($z_i = 0$), our derived volumetric ratios above ($V/Vo$) would imply gas expansion taking place in the uppermost tens of metres of the conduit (60 and 45 m for the 3 July and 28 August explosions, respectively). However, since gas expansion is the driver of magma column deformation, the expansion must occur deeper than the portion of the conduit affected by the ground inflation, implying that solutions with $z_o < 240$–380 m (our volumetric expansion source) are not considered. The "shallowest" possible solution thus corresponds to expansion starting immediately at the base of the deformation source ($z_o = 380$ m and 240 m for the 3 July and 28 August explosions, respectively), in which case we find that the fragmentation ($z_i$) occurs at the same shallow depth for both eruptions (28 m and 16 m below the

surface on 3 July and 28 August, respectively). Alternatively, if the fragmentation depth is fixed at 150 m below the vent, as inferred from the time delay between the thermal and acoustic onsets for the 15 March 2007 paroxysm[29], the onset of the expansion process is constrained deeper in the system (2430 m and 1845 m below the craters for the 3 July and 28 August events, respectively). A 1845–2430 m depth range matches the deepest source depths of gas slugs at Stromboli[45,46]. The ascent and degassing of gas-rich magma likely begins far deeper at Stromboli[33,34,45,47], perhaps from a deep magma storage zone as deep as ~6 km below sea level[34]. We argue, however, that initial magma ascent beginning at such depth may produce a deformation signal too small to be detected by our instruments at the surface. We also infer that a deeper expansion onset would translate into a high ascent velocity (>35 m/s) for the vesiculated magma before its fragmentation (Eq. 2). Our calculations instead suggest ascent rates for the expanding gas-magma mixture between 15.3 m/s (3-July event) and 9.5 m/s (28 August). These ascent rates are within the same order of magnitude of those estimated from physical modelling[47], mineral textures[48], volatile contents in natural glass embayments[34], and decompression experiments[49,50]. Similarly, high rates of magma ascent were reported for other basaltic systems, such as the 1974 eruption of the Volcán de Fuego (8–21 m/s) (ref. [51]).

We use the systematic (scale invariant) exponential rate at which the ground inflates before an explosion (of any size) to develop an early warning alert system that automatically recognizes the deformation pattern preceding a paroxysm (Supplementary Video 1). The detection algorithm is based on a pattern matching analysis between the observed tilt signal and a template represented by the first 350 s long theoretical inflation rate induced by gas expansion in a viscous fluid. The recorded deformation pattern, $\tau(t)$, is considered to match the template when the best fit is >0.85 and the amplitude of deformation is >0.3 µrad (Methods and Supplementary Fig. 2). When these two criteria are applied to the ground deformation records streamed by tiltmeters over the last 14 years (from 2006 to 2020), the three paroxysms of March 2007, July 2019 and August 2019 are automatically detected 4–5 min before the onset of the explosions, and neither negative nor false positive alerts are issued in the same analysed time period (Supplementary Fig. 3). On 28 August 2019, although the early warning alert system was still being tested, it allowed us to warn the Italian Civil Protection at ~10:12 GMT, 5 min before the paroxysm and ~9 min before a tsunami (which was triggered by the pyroclastic material laterally ejected towards the Sciara) struck the coast of Stromboli (see Supplementary Video 1). To the best of our knowledge, this is the first example of a warning issued to authorities shortly before a volcanic explosive eruption occurred and to the population before a tsunami was generated.

Sudden, violent explosive blasts often occur at otherwise "calm" open-vent volcanoes without being preceded by detectable changes in the monitored parameters, hampering our ability to assess the risk associated with these phenomena. At Stromboli, explosions span almost seven orders of magnitude in terms of the erupted volumes (Fig. 3b), but their associated ground deformation obeys the same inflation rate. Ground tilt records show how the rate of the inflation is a common characteristic for the full wide range of explosive activity and that this conduit process remains the same but operates at different scales (Figs. 2b and 4a). The amplitude and duration of the pre-blast ground inflation scale with the magnitude of the explosive eruption (Figs. 2b and 3b). In the case of paroxysms, inflation starts almost 10 minutes before the explosive onset (Figs. 2b and 5; Supplementary Video 1). We conclude that, in spite of different magmas being involved, the explosive dynamics at Stromboli follow a self-similar

ground deformation process that repeats, though at different scales, with the same exponential rate of inflation. We explain these ground inflations as generated by the pressure increase on the conduit wall induced by rapid volumetric expansion of the gas in a highly vesiculated magma batch. Our inferred shallow pressure source (between 370 and 450 m a.s.l.) coincides with the location of the tensile failure inferred to occur in a crystal-rich magma mush to explain regular explosive activity[42]. Overpressuring of the conduit magma and the consequent upward migration of the magma column may well explain the magma overflow events from the summit craters (or the increasing magma effusion rates from lateral vents) repeatedly observed prior to Stromboli's paroxysms, such as in the minutes before the March 2007 (ref. [52]) and July 2019 events, which we interpret as the magma column being squeezed outside the vent by the expansion of the gas-rich magma batch.

Our results offer a new perspective for monitoring volcanoes. Applications of this pattern matching methodology can overcome caveats in using raw tilt signals in real time, which have long prevented their use as effective monitoring tools. The scale-invariant, persistent exponential rate of deformation provides a robust statistical basis for an early warning alert system grounded in a quasi-deterministic approach. Thus, ground deformation opens new avenues to image volcanic processes at different time scales, and we foresee further improvements of the actual early warning alert system through the integration of other monitoring techniques in the near future.

## Methods

**Instrumental network**. The monitoring network was deployed in January 2003 (ref. [53]) and has since been expanded. At the time of the 2019 paroxysms, 5 broadband seismic stations, 10 acoustic sensors, 3 tiltmeters, 2 thermal cameras, 1 visible camera, 2 UV cameras, and 1 multigas sensor were fully operational, and their data were processed in real time (Supplementary Fig. 1). Data provided by the network are integrated and interpreted to define the explosive process step by step at a one second resolution. Ground deformation is recorded by using 2 borehole tiltmeters (Pinnacle series T5000) with a sensitivity of 1 nrad at a 1 Hz sampling rate that are installed at a depth of ~6 m. Each seismic station is equipped with a Guralp CMG-40T broadband sensor (natural period: 30 s) sampled at 100 Hz and 24 bits.

**Source localisation by an open-conduit deformation model**. The depth of the source of ground inflation is estimated by integrating the tiltmeter signal recorded at station OHO (Supplementary Fig. 1) with the tilt component derived by broadband seismometers following the method developed in previous studies[12,13]. All the stations are located less than 1000 m from the vent and show clear inflation of the ground consistent with almost isotropic expansion centred at the summit crater (Fig. 1). We explain the pressure increase at the shallow portion of the conduit walls as induced by the expansion of the highly vesiculated magma during its rise towards the surface (Fig. 5). Following an open-conduit deformation model[36], the radial tilt, t, is related to the overpressure, DP, in a cylindrical pipe with radius a by the following equation:

$$\tau = G \frac{a^2}{\mu} \Delta P \tag{5}$$

where $\mu$ is the rigidity modulus and $G$ is the Green's function relative to the station at a distance $x$ from the conduit (Fig. 5):

$$G = \frac{x}{2(h_2 - h_1)} \left[ \frac{3h_2^2}{R_2^5} - \frac{2\nu}{R_2^3}(h_2 - h_1) + \frac{h_2}{R_2^3} - \frac{h_1}{R_1^3} - (2\nu - 1)\left( \frac{1}{R_2(R_2 + z_2)} - \frac{1}{R_1(R_1 + z_1)} \right) \right] \tag{6}$$

where $h_1$ and $h_2$ are the top and bottom depths, respectively, of the portion of the conduit that generates the ground deformation and $R_1 = \left(x^2 + h_1^2\right)^{\frac{1}{2}}$ and $R_2 = \left(x^2 + h_2^2\right)^{\frac{1}{2}}$ are the associated distances between the station and the portion of the conduit representing the source (Fig. 5a).

For an ideal Poissonian solid (Poisson modulus of 0.25), the overpressure, $\Delta P$, can be related to the volume change, $\Delta V$, of the source volume $V$, by:

$$\Delta P = \frac{5}{3}\mu \frac{\Delta V}{V} \tag{7}$$

Then, for a cylindrical source of length $L = h2 - h1$ and radius a, $(V = \pi a^2 L)$, combining Eqs. (5) and (7), we obtain:

$$\tau = \frac{5}{3\pi L} G \Delta V \tag{8}$$

We apply a search algorithm for the best conduit source position assuming that h1 can vary with a step of 10 m from 0 to 10,000 m and that L can vary from 10 to 1000 m below the surface. The best solution is thus estimated through a least square minimization among all the 99,000 possible solutions between the ratios of the Green's functions, Gi/Go, and the measured tilt, $\tau_1/\tau_o$, at station $i$, where $Go$ and $\tau_o$ are relative to the reference station. Once the best source position is found, the volumetric change, $DV$, is calculated using Eq. (8).

We caution that our open-conduit deformation model does not account for the effects of topography on the measured tilt signal. However, finite element model (FEM) accounting for topographic effects returned very comparable results, suggesting that while topography may have a large effect on the tilt direction, it plays a minor role (<15%) in the tilt amplitude.

**Early warning algorithm**. The ground tilt is sensitive to a series of unpredictable natural sources of noise due mostly to atmospheric pressure, ground temperature, meteoric precipitation, earthquakes, and deformations resulting from magma intrusion in addition to deformations induced by artificial sources[13]. These sources can generate spurious ground tilt signals that can eventually mask the deformation pattern associated with the explosive process. The temporal patterns of these non-explosion-related, undesired deformation signals are unpredictable and cannot be filtered out a priori from the recorded tilt in real time. To prevent contamination of the deformation pattern, neither filtering nor pre-processing is applied to the recorded signal before the best fit is searched. The tilt amplitude in a 350 s long time window is simply normalized before it is compared to a template (inset in Supplementary Fig. 2), represented by the normalized theoretical ground inflation. Real-time pattern matching analysis is applied every second in a 350 s long backward window, $tw$:

$$R(t) = 1 - \left\{ \sum_{ts=t-tw}^{t} \left[ \tau(t_s) - \tau_{ref}(t_s) \right]^2 \Big/ \sum_{ts=t-tw}^{t} \left[ \tau_{ref}(t_s) \right]^2 \right\}^{\frac{1}{2}} \tag{9}$$

An early warning is triggered when the best fit $R(t)$ is >0.85 and the tilt variation in the considered time window is >0.3 µrad. Testing of this early warning algorithm using tilt records over the past 14 years (from 2006 to 2020) shows (Supplementary Fig. 4) that the three paroxysms of 15 March 2007, 3 July 2019, and 28 August 2019 are automatically detected almost 5 min before the onset of the eruption with no false alerts. On the basis of this test, at present, an early warning alert system is fully operational at Stromboli and is linked to an acoustic alert network for the population.

## Data availability

All data generated or analysed during this study are included in this published article (and its Supplementary Materials) or available from the corresponding author upon reasonable request.

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

## Acknowledgements

This work was supported by the Italian Civil Protection in the framework of the DEVNET project.

## Author contributions

M.R. and G.L. conceived the early warning alert system. M.R., M.P., D.D.D. and A.A. wrote the manuscript. M.R. and T.N. conceived the conduit model. All authors contributed in the data collection, instrument setup and maintenance. All authors analysed the data and contributed to finalizing the manuscript. D.M. and A.R. helped integrate the warning procedure into the Italian Civil Protection system. M.R. supervised the project and the collaborations.

## Competing interests

The authors declare no competing interests.

**Additional information**

