## [Peer Review File · Nature Communications]

REVIEWER COMMENTS

Reviewer #1 (Remarks to the Author):

The study shows a simple analysis and modeling of a predictable unrest signal at Stromboli volcano. The authors show evidence that the quasi-exponential unrest is scale invariant, spanning many orders of magnitude and that this results from the ascent and expansion of magma, regardless whether it is either gas rich or gas poor. In general I consider the article well written, the data provide support for the conclusions and the science is relevant and state of the art with respect to the current questions in volcanology. I have two major comments. The first one, is it possible that the ascent of the gas slug does not result in conduit shearing that can also be detected by the tilt network? The second comment is that there is not enough detail on the formulas used and how they relate to the different model parameters. For example, where does the V/V_0 ratio come from. This is straightforward to address. I recommend the article for publication after addressing these minor revisions.

33: at Stromboli, since the model is probably not applicable to other volcanoes with lack of tilt data.

27-29: Actually the model does not take into account the difference in crystal and gas content of the lava erupted in the paroxysms and minor eruptions so this phrase must be improved.

63: What is volatile poor? Please clarify.

79-80: is it possible to detect the signals with 1 Hz GPS data? Is there GPS data at Stromboli? A few lines about this would be beneficial for the article to highlight the limitations of the seismic approach and for authors with a more geodetic rather than seismological background

88: "than the estimated for ..."

131: intensity is a better word than violence for this sentence.

133-138: I'd add the disclaimer that even though this model is supported by other data sets, the amount of gas and the differences between the more crystal rich and crystal poor basaltic magmas are not explicitly taken into account in the model.

141: is it possible that the magma ascent also produces conduit shearing and not only conduit pressurization?

151: where is this formula coming from?

157: the equation does not include the V/V_0 ratio, so where does this come from?

173-174: is this from some of the equations?

183: "the onset of the expansion ..."

181-184: is this supported by the tilt data?

175-177: is this inference taken from some other study? Because the model does not include a fragmentation threshold.

177: "that are not considered".

180: "occurs at the same shallow depth"

187-188: why is this deformation source not detectable? Is it because the volume change is too small for the source depth?

217: "the same inflation rate".

277: where is this formula from?

282: this is confusing. Eq(1) and Eq(3) do not include either L or pi, so I don't understand where formula 8 comes from.

488: based on the temperature. The scale of the intensity of the eruption can also be measured based on the RSAM or some other earthquake statistics.

497: cite reference for the method.

516: how sure are you that the ascent of magma results in conduit pressurization at the bottom of the magma column and not in shearing of the conduit walls?

Line 517: "on top".

Signed

Francisco Delgado, IPGP

Reviewer #2 (Remarks to the Author):

SUMMARY

In this manuscript, the authors present the results from the analysis of ground deformation data at Stromboli volcano, Italy to 1. identify short-term precursors to sudden large eruptions and 2. implement this technique as a near-real time early warning system.

The science is solid, the manuscript very well written, and the figures relevant and of suitable quality. The results from this work are an important milestone for our field and volcano monitoring: they bridge science and operations to add immediate value to society, which is incredibly vital.

This manuscript will be, in my opinion, an excellent and very well-suited contribution to Nature Communications after only a few very minor edits (see below and attached).

COMMENTS

My comments are mostly editorial, minor in nature, and can be found in the attached annotated manuscript.

Only a small - yet important - comment about the title:

Put simply, the deformation signal is "simply" a late symptom of magma rise/vesiculation that leads to eruptions. So this signal does not provide direct information about the actual trigger for major and paroxysmal eruptions: those occur at greater depths. The title and scope of the paper should therefore reflect this. I must emphasise that this does not make this work any less impactful: these results are an important step forward in terms of monitoring - we just want to be accurate in the title and statements.

Interestingly, from an implementation point of view, one of the take home messages for me was that it wasn't possible to filter out non-volcanic (and false positive) signals on the tilt timeseries. As a result, the authors resorted to pattern matching between near-real time streaming data and previous events to detect previously recognise anomalies preceding impending large eruptions. This is quite a crucial point since volcano observatories rarely use tiltmeter data in near real time, due to the ineffectiveness of the data filtering to remove "unwanted" signal. Beside the great science, this work is therefore also breaking new ground from an operations perspective, which is fantastic. Maybe the authors would like to develop this a little more in the latter part of the manuscript if there is space and appetite.

The attached manuscript is quite substantially annotated and will hopefully assist the authors in improving the readability of what is already an excellent article.

CONCLUSIONS

I strongly support the publication of this manuscript in Nature Communications. The science is solid, the results very well presented, and their impact on society (as an warning system) important and clearly laid out.

I remain available to the authors and the editor should any clarifications on the review be needed.

Kind regards,

Nico

Nico Fournier
GNS Science
Taupo
Aoteraroa New Zealand
n.fournier@gns.cri.nz

Reviewer #3 (Remarks to the Author):

The work presented in this paper can be separated in three parts pertaining to 1) ground deformation data collected over almost 20 years at Stromboli Volcano and the localization of its source region, 2) the development of a physical model explaining the observations, especially their scale-invariance, and 3) the application of this knowledge to set up an operational early warning system at this island. It is therefore a good illustration of the whole scientific process of data collection, formulating a hypothesis and model, and finally the practical application of the knowledge obtained.

Strombolian activity has often been associated with the rise of pressurized gas slugs in the conduit. While this model is quite successful in describing the observed dynamics, over the last few years the “magma foam” model, i.e. the rise of a batch of vesiculated magma, has gained support, especially to explain ash generation through fragmentation. The present study is, to my knowledge, one of the first to actually formulate a physical model for this process and to apply it successfully to observational data.

In the light of the two paroxysmal eruptions at Stromboli in summer 2019 and the ongoing discussion about reopening its summit for touristic excursions, the work presented also possesses relevance beyond the strict scope of field.

The localization of the deformation sources from the deformation data is sound using a basic cylindrical open conduit model and a minimum of assumptions. The assumptions made are reasonable and supported by literature. In this section though, I miss a more explicit reference to the influence of Stromboli’s complex topography and geology on the comparability of the stations (around line 258).

The conclusion that the basic process within the conduit is the same for eruptions over several orders of magnitude is, in my opinion, valid and supported by the presented results. From a purely statistical standpoint, the different sample sizes in the three eruption classes (>10.000 regular, 39 major, 4 paroxysmal eruptions) may make a comparison between the different scales seem problematic. Nevertheless, the excellent correlations of the signal shapes (Fig. 2 and 4a) and also between erupted volume and tilt amplitude (Fig. 3) in combination with the model explanation developed warrant the conclusion. The correlation coefficients and the precision of the results are quoted where required, allowing for an independent assessment by the reader.

The model developed for a batch of gas-rich magma expanding during ascent in the conduit, is simple in the positive sense that it only makes basic and well supported assumptions and still successfully offers an explanation for the observations. More complex models are conceivable, e.g. with a non-Newtonian depth-variable magma rheology accounting for variances in chemistry and temperature within the magma column, but experience with the slug model mentioned above has shown, that a similarly basic model can already capture the majority of the physics involved.

The train of arguments to frame the depth of the onset of gas expansion to 1800–2400 m (lines 172–

195) is well described and supported. Even at the bare minimum, it demonstrates that the model developed is not in contradiction to any observational evidence.

The early warning algorithm demonstrates how field observations and model building can lead to practical application. Again it is characterized by its conceptual simplicity but successful applicability. The detection rate and lack of false positives on “hindcasting” the last 14 years of activity, as well as the successful early warning issued shortly before the latest paroxysm, support the approach. A complete assessment of the method will however only be possible after observing it in operation in the future, especially regarding false negatives, i.e. paroxysms not captured in advance.

In summary, the methodology and conclusions drawn are well supported by literature citations and the observational data. All the necessary information is provided to enable the reader to make his own assessment of the results. It seems well possible to reproduce the work flow given the information contained in the paper.

However, I would like to make one suggestion in order to improve how the reader is guided through the manuscript: When entering the Discussion, it did not immediately become clear to me, how the model developed is connected to the data evaluation described before (line 143–144). In particular, what quantity and why is compared to the tilt data (lines 161–171). As I understand it, tilt τ is proportional to pressure ΔP according to eq. 1, which in turn is proportional to volume ΔV assuming an ideal gas, being proportional to depth z according to the model. The temporal evolution $\tau(t)$ and $z(t)$ can thus be correlated. I would suggest a short explanation of the connection between data and model at around either lines 144 or 161.

So, in conclusion, the manuscript is generally well written, the methodology sound and all the information is present to allow the reader to reproduce the work and to assess its uncertainties. It is relevant and current as it properly introduces a physical model for expanding vesiculated magma driving Strombolian activity to the field and in addition describes a promising operational early warning system theoretically based on said model. I strongly support the publication of this work, after my suggestions above have been considered.

Best regards,
Jost von der Lieth

Minor issues:

Line 144 but pertaining to the entire manuscript: The deformation is described as “almost exponential” only here. To be very pedantic, neither the data nor the model is exactly exponential. Nevertheless, if taken in the broader sense, “exponential” is the appropriate adjective for the deformation trend throughout the paper.

Line 88: what _was/is_ estimated

Line 126: limit reference to Fig. 4a

Line 165: limit reference to Fig. 4b

Line 248: define step-by-step _the_ explosive process

Line 279: check references to eqs. (5) and (7)

Fig. 2: red underlining

Sup., end of 1st paragraph: During this period _of_

Point by point response to REVIEWERS' comments

Reviewer #1 (F. Delgado):

I have two major comments. The first one, is it possible that the ascent of the gas slug does not result in conduit shearing that can also be detected by the tilt network? The second comment is that there is not enough detail on the formulas used and how they relate to the different model parameters. For example, where does the V/V_0 ratio come from. This is straightforward to address. I recommend the article for publication after addressing these minor revisions.

We thank Dr Delgado for the positive assessment. Regarding the two specific comments, we now state in the revised text that “*Shear stress generated by the magma ascent is negligibly small (Nishimura 2009)*”. We also modified the text to better clarify the equations and the relationship between the different parameters in the model.

33: at Stromboli, since the model is probably not applicable to other volcanoes with lack of tilt data.

Done

27-29: Actually, the model does not take into account the difference in crystal and gas content of the lava erupted in the paroxysms and minor eruptions so this phrase must be improved.

Done: the sentence was rephrased to avoid reference to difference in crystallinity. Although we do not model the difference in gas content between the two magmas, the high gas content of the ascending magma (well supported by petrology) is indirectly implied by the equations (gas is the driver of magma expansion in the conduit).

63: What is volatile poor? Please clarify.

Done

79-80: is it possible to detect the signals with 1 Hz GPS data? Is there GPS data at Stromboli? A few lines about this would be beneficial for the article to highlight the limitations of the seismic approach and for authors with a more geodetic rather than seismological background

1Hz GPS data are available for the 5 April 2003 paroxysm. This is now mentioned at lines 105,

where we also inserted a new reference. Since then, however, 1 Hz GPS data have not been available at Stromboli.

88: “than the estimated for ...”

Done

131: intensity is a better word than violence for this sentence.

Done

133-138: I’d add the disclaimer that even though this model is supported by other data sets, the amount of gas and the differences between the more crystal rich and crystal poor basaltic magmas are not explicitly taken into account in the model.

Done

141: is it possible that the magma ascent also produces conduit shearing and not only conduit pressurization?

We now state in the revised text that “Shear stress caused by the upward magma migration has been found to only account for ~16% of volcano deformation (Nishimura, 2009) and is thus neglected”

151: where is this formula coming from?

We now explain in the revised text that equation (3) is derived from rearranging the perfect gas law ($PV = \text{const}$, or $P_0V_0 = P_1V_1$). We now state that “Assuming ideal gas behavior and that gas expansion starts at depth z_0 , the perfect gas law (...) can be rearranged to demonstrate that the initial gas volume (...) in the magma batch will grow as function of the depth $z(t)$ upon ascent.”

157: the equation does not include the V/V_0 ratio, so where does this come from?

The ratio V/V_0 is equivalent to the ratio R/R_0 found with the best fit between data and model. We preferred to express the solutions of our modelling in terms of V/V_0 ratio rather than R/R_0 for the direct proportionality between volumetric (V) expansion and depth (z). Note the perfect gas law can be also re-written as $V_0z_0 = V_1z_1$. We added the following text for clarification “It is noteworthy that although model equations (eq. 2-3) are written in terms of the R_g/R_0 ratio, we rather prefer expressing our solutions in terms of the equivalent volumetric ratio (V/V_0).”

173-174: is this from some of the equations?

Done

183: “the onset of the expansion ...”

Done

181-184: is this supported by the tilt data?

No, this is now clarified in the text.

175-177: is this inference taken from some other study? Because the model does not include a fragmentation threshold.

We have specified in the text these thresholds derive from our source modeling that confines the source of deformation at 240-380 m depth in the conduit.

177: “that are not considered”.

Done

180: “occurs at the same shallow depth”

Done

187-188: why is this deformation source not detectable? Is it because the volume change is too small for the source depth?

Clarified in the text: *”We argue, however, that such initial magma ascent may produce a deformation signal too small to be detected by our instruments at the surface”.*

217: “the same inflation rate”.

Done

277: where is this formula from?

Equation 7 comes from elasticity theory ($P=K\theta$, where K is the bulk modulus and θ is the volumetric strain $\Delta V/V$) and it represents the relation between pressure and volumetric strain for ideal Poissonian solid. The new text reads as *“For an ideal Poissonian solid...”*

282: this is confusing. Eq(1) and Eq(3) do not include either L or π , so I don't understand where formula 8 comes from.

This is correct. There was a typo in the equation numbers, which has now been corrected.

488: based on the temperature. The scale of the intensity of the eruption can also be measured based on the RSAM or some other earthquake statistics.

Done

497: cite reference for the method.

Done

516: how sure are you that the ascent of magma results in conduit pressurization at the bottom of the magma column and not in shearing of the conduit walls?

see reply above. This is explained in Nishimura (2009) and Kawaguchi and Nishimura (2015)

Line 517: “on top”.

Done

Reviewer #2 (N. Fournier):

Put simply, the deformation signal is "simply" a late symptom of magma rise/vesiculation that leads to eruptions. So this signal does not provide direct information about the actual trigger for major and paroxysmal eruptions: those occur at greater depths. The title and scope of the paper should therefore reflect this. I must emphasise that this does not make this work any less impactful: these results are an important step forward in terms of monitoring - we just want to be accurate in the title and statements.

We thank the reviewer for this suggestion. The title has been changed to accommodate this comment.

Interestingly, from an implementation point of view, one of the take home messages for me was that it wasn't possible to filter out non-volcanic (and false positive) signals on the tilt timeseries. As a result, the authors resorted to pattern matching between near-real time streaming data and previous events to detect previously recognise anomalies preceding impending large eruptions. This is quite a crucial point since volcano observatories rarely use tiltmeter data in near real time, due to the ineffectiveness of the data filtering to remove "unwanted" signal. Beside the great science, this work is therefore also breaking new ground from an operations perspective, which is fantastic. Maybe the authors would like to develop this a little more in the latter part of the manuscript if there is space and appetite.

We are very pleased by this comment. Referee is correct, filtering is ineffective to remove unwanted signals. Pattern matching seems to be at the moment the easy and effective way of dealing with tilt signal. Following the suggestion we spend few words to stress this point on the manuscript: *“Our results open new perspective in the way we can monitor volcanoes. Application of the pattern matching methodology overcomes caveats in using raw tilt signals in real-time, which have long prevented their use as effective monitoring tools.”*

The attached manuscript is quite substantially annotated and will hopefully assist the authors in improving the readability of what is already an excellent article.

All editing requests in the provided PDF were accepted.

Line 32, rephrase, what is the take away point here?

The sentence was rephrased as requested.

Lines 44-45 I am not sure the limitation is our modelling. It seems to be more related to being unable to detect and recognize such signal in near-enough real time that they can be used as early warning. Maybe rephrase? this could probably be illustrated easily by showing a long time series (but there is no need here)

The reviewer is correct, as modeling is less important, limitations are less substantial in this

specific case. The sentence has been rephrased as requested to focus on the measurement challenges.

Line 83 at the ground surface? Clarify

Yes, clarified in the text.

Lines 131-134. doesn't the magnitude of deformation vary with scale of explosions? please rephrase and clarify

Agreed, text clarified.

Line 159. how does this vary with volatile content? (eg strombolian vs more explosive)

The volatile content is accounted for in the model by the term R_0 (radius of the initial gas bubble) value. However, we show (eq. 4) for that the inflation rate does not depend on the volatile content but rather on the z_0/R_0^2 ratio.

Line 173. what is the uncertainty? Is it really different from 610s?

We agree with the two reviewers. There is no much difference between the two solutions which are in fact identical within uncertainty. However, 590 and 610 represent the solutions with the best fit with the tilt signal. The text was revised to accommodate this “*A similar (within uncertainty) solution (expansion time t_b of 590 s for a volumetric expansion V/V_0 of 12) is derived for the 28 August 2019 eruption (best fit of 0.98)*”

Line 190. Rev#2 accuracy? uncertainty? round instead?

Yes, rounded as requested

Line 194-195 explain why. limited network aperture and/or instrumental limitations?

Clarified in the text: “*We argue, however, that such initial magma ascent may produce a deformation signal too small to be detected by our instruments at the surface*”.

Line 224-225. the pattern is similar. But am I mistaken or the rates look different in Fig. 2?

No, we don't see any obvious difference in the rates

Reviewer #3 (J. von der Lieth):

All the textural comments and request for changes made by the reviewer in the marked manuscript have been addressed

The localization of the deformation sources from the deformation data is sound using a basic cylindrical open conduit model and a minimum of assumptions. The assumptions made are reasonable and supported by literature. In this section though, I miss a more explicit reference to the influence of Stromboli's complex topography and geology on the comparability of the stations (around line 258).

We have followed the reviewer's comment and have added the following text on this argument *"We caution that our open conduit deformation model does not account for topography effects on the measured tilt signal(s). However, Finite Element Modeling (FEM) accounting for topography shows that while topography may have a large effect on tilt direction, it plays minor role (<15%) on tilt amplitude"*.

However, I would like to make one suggestion in order to improve how the reader is guided through the manuscript: When entering the Discussion, it did not immediately become clear to me, how the model developed is connected to the data evaluation described before (line 143–144). In particular, what quantity and why is compared to the tilt data (lines 161–171). As I understand it, tilt τ is proportional to pressure ΔP according to eq. 1, which in turn is proportional to volume ΔV assuming an ideal gas, being proportional to depth z according to the model. The temporal evolution $\tau(t)$ of and $z(t)$ can thus be correlated. I would suggest a short explanation of the connection between data and model at around either lines 144 or 161.

The relation between tilt τ , pressure ΔP and volumetric change ΔV described by the Reviewer is correct. However, the volumetric expansion model (equation 2) is used here only to explain the origin of the exponential rate of the over pressure increase within the conduit. There is no physical relation between the V/V_0 ratio during the gas expansion and the volumetric dilatation ΔV in equation (8) which depends on the elastic properties of the surrounding rocks. We think this is now better defined in the text as follow: *"The rate at which ground deforms is thus controlled by ascent velocity of the expanding gas-rich magma batch into the conduit (equation 2).*

We have fully accepted the invitation of the reviewer to state more explicitly what model equation is fitted with what observation. This is now made in the revised discussion: *"Assuming the expansion time t_b coincides with the total duration of tilt inflation (windowed from 540 to 720 s, from the duration of the pre-explosion inflation; Figs. 1 and 5), we fit the normalized tilt patterns with the normalized volumetric expansion ratio V/V_0 ."*

Minor issues:

Line 144 but pertaining to the entire manuscript: The deformation is described as “almost exponential” only here. To be very pedantic, neither the data nor the model is exactly exponential. Nevertheless, if taken in the broader sense, “exponential” is the appropriate adjective for the deformation trend throughout the paper.

Done

Line 88: what _was/is_ estimated

Done

Line 126: limit reference to Fig. 4a

Done

Line 165: limit reference to Fig. 4b

Done

Line 248: define step-by-step _the_ explosive process

Done

Line 279: check references to eqs. (5) and (7)

Done

Fig. 2: red underlining

Done

Sup., end of 1st paragraph: During this period _of_

Done

REVIEWERS' COMMENTS

Reviewer #1 (Remarks to the Author):

The manuscript has been significantly improved and all the points where I raised some remarks were properly addressed.

The only thing that I would like to add is that in lines 180-185 the formula that is fit to the tilt time series are equations 8, 3 and 4, if I understood correctly the manuscript.

After this I recommend the manuscript for submission.

Reviewer #3 (Remarks to the Author):

The authors fully addressed my own and, as far as I can judge, the other reviewers' comments and suggestions. I support all the small revisions made to the manuscript. What was a generally well written and technically sound study already, has mainly improved in readability and the ability to communicate its findings.

There is nothing more I can add to my first report except that I now recommend the manuscript for publication without any further changes.

Best regards,
Jost v. d. Lieth

One minor typo on line 257: provides

Reviewer #1 (F. Delgado):

Reviewer #1 (Remarks to the Author):

The manuscript has been significantly improved and all the points where I raised some remarks were properly addressed.

The only thing that I would like to add is that in lines 180-185 the formula that is fit to the tilt time series are equations 8, 3 and 4, if I understood correctly the manuscript.

After this I recommend the manuscript for submission.

We thank Dr Delgado for his second review. We have rephased the sentence to further clarify this point.

Reviewer #3 (J. von der Lieth):

The authors fully addressed my own and, as far as I can judge, the other reviewers' comments and suggestions. I support all the small revisions made to the manuscript. What was a generally well written and technically sound study already, has mainly improved in readability and the ability to communicate its findings.

There is nothing more I can add to my first report except that I now recommend the manuscript for publication without any further changes.

Best regards,
Jost v. d. Lieth

One minor typo on line 257: provides

We thank Dr v. d. Lieth for his second review. We have corrected the typo in line 257.